Initial decomposition of floating leaf blades of Nymphoides peltata (S.G. Gmel.) O. Kuntze (Menyanthaceae): causes, impact and succession

Klok Peter F. 1 2
van der Velde Gerard 1 3 g.vandervelde@science.ru.nl
1 Animal Ecology and Physiology, Radboud University Nijmegen , Nijmegen , Netherlands
2 Department of Particle Physics, Institute for Mathematics, Astrophysics and Particle Physics, Radboud University Nijmegen , Nijmegen , Netherlands
3 Naturalis Biodiversity Center , Leiden , Netherlands
Daehler Curtis
Electronic publication date: 2023 Dec 21
Publication date: 2023
Volume: 11
Electronic Location ID: e16689
Received 2023 May 10; Accepted 2023 Nov 27
Copyright: © 2023 Klok and van der Velde
Copyright year: 2023
Copyright holder: Klok and van der Velde
License: This is an open access article distributed under the terms of the Creative Commons Attribution License, which permits unrestricted use, distribution, reproduction and adaptation in any medium and for any purpose provided that it is properly attributed. For attribution, the original author(s), title, publication source (PeerJ) and either DOI or URL of the article must be cited.
License URL: https://creativecommons.org/licenses/by/4.0/

Keywords: Enclosure, Herbivores, Laminae, Mesocosm, Nymphaeid growth form, Senescence, Succession decomposition causes

Funding: The authors received no funding for this work.

==============================
Background

During a study on the outdoor floating leaf blade production of Nymphoides peltata (S.G. Gmel.) O. Kuntze (Fringed Water Lily), initial leaf blade decomposition was studied by simultaneously measuring infected, damaged and lost area of floating leaf blades.

Methods

Data on initial decomposition over time were collected for all leaves during one growth season in four plots: two in outdoor mesocosms and two in an oxbow lake. Each leaf was tagged uniquely upon appearance in a plot. The vegetation in the mesocosms differed with respect to plant species, one contained a monoculture of N. peltata and the other N. peltata associated with Glyceria fluitans (L.) R. Br. and G. maxima (Hartm.) Holmb. The lake plots were situated within a monospecific N. peltata stand, differing in depth and position within the stand. Leaf length, visually estimated percentages of damaged area for each damage type, and decay of the tagged leaves were recorded bi-weekly. When the leaf blades sunk under the water surface or disappeared completely, they were no longer followed. Under water the leaves decayed and were consumed by snails completely, so contributing to the detritus food chain.

Results

The observed causes of damage on floating leaves were consumption and/or damage by waterbirds (Fulica atra), pond snails, caterpillars (Elophila nymphaeata, Cataclysta lemnata), chironomid larvae (Cricotopus trifasciatus), infection by a phytopathogenic fungus (Septoria villarsiae), senescence by autolysis, and microbial decay. Successional changes in causes of leaf decomposition and impacts of different causes are discussed.

Introduction

Aquatic macrophytes can be considered as the basic frame of wetland ecosystems, called macrophyte-dominated systems (Den Hartog & Van der Velde, 1988; Jeppesen, Søndergaard & Christoffersen, 1998). The nymphaeid macrophyte, a growth form represented by species of Nymphaeaceae, Menyanthaceae, Potamogetonaceae, Polygonaceae and Aponogetonaceae, forms the base of nymphaeid-dominated systems (Van der Velde, 1980). This growth form is characterized by floating leaves, flowers on or elevated above the water surface and roots in the sediment of shallow open waters or littoral borders (Luther, 1983; Van der Velde, 1981).

In shallow alkaline freshwater lakes, Nymphoides peltata can cover large areas (e.g., Brock, Van der Velde & Van de Steeg, 1987). Information about distribution, growth form, importance of floating leaves, habitat and environmental conditions of N. peltata (Fig. 1) is presented in Klok & Van der Velde (2022).

Figure 1 Floating leaf blades of Nymphoides peltata in spring.

Decomposition of floating leaf blades

Decomposition of leaves in natural conditions (Fig. 2) involves a complex set of interacting processes, which can be roughly classified into internal (physiological) and external (abiotic or biotic) processes (Van der Velde et al., 1982; Kok, 1993). At first only host specific species—more or less specialized and often restricted to particular plant taxa—are able to break through the defense system to consume fresh plant tissue. At a later stage, when the defense system has been weakened, other species colonize the leaves.

Figure 2 Successive stages of decomposition of floating Nymphoides peltata leaf blades.

(A–F) Leaves proceeding towards senescence, (G and H) autolysis, (I and J) autolysis and microbial decay, (K) microbial decay and (L) fragmentation.

Decomposition of young leaf blades of N. peltata already starts under water before they unroll with consumption and/or damage. This is visible when they enroll at the water surface often as a row of similar damage patterns on the same leaf blade (Klok & Van der Velde, 2019). During initial decomposition, when leaves are still floating, macrophyte tissue is used by herbivores and by phytopathogenic and saprotrophic microorganisms (Kok, 1993). Various causes and stages of decomposition can be recognized on a single leaf. Damage of leaves can induce the leaching of soluble carbohydrates such as oligosaccharides and starch, proteinaceous and phenolic compounds, some of which can be rapidly metabolized by microorganisms. Finally, plant tissue senesces by autolysis, coloring yellow by the resorption of chlorophyll (Lammens & Van der Velde, 1978; Kok, 1993), followed by further decomposition initiated by weak pathogens and facultative herbivores, leading to the production of debris and fecal pellets (Kok, 1993). The chemical composition of plant tissue also changes during autolysis due to hydrolysis of macromolecules (which may weaken tissue structure), resorption of nutrients (like N and P, as well as carbon compounds such as starch) and loss of secondary compounds (Kok, 1993). Furthermore, leaves are colonized by microorganisms, causing microbial enrichment which makes the tissue more attractive for detritivorous macroinvertebrates (Kok, 1993). Decayed floating leaves sink underwater to the bottom, where they provide a resource for detritus-based benthic food webs during further decomposition (Klok & Van der Velde, 2019). To study the latter decomposition process, the litterbag method is commonly used (Brock et al., 1982; Brock, 1985; Wieder & Lang, 1982). As N. peltata only grows in alkaline water (Van der Velde, Custers & De Lyon, 1986; Smits et al., 1988) and development of floating leaves is dependent on calcium uptake of this plant, decomposition is studied only in alkaline water (Smits, Schmitz & Van der Velde, 1992).

Research questions

The present study covers initial decomposition of floating leaves of Nymphoides peltata (S.G. Gmel.) O. Kuntze in different water bodies by investigating the effects of damage causes found on floating leaves. In an earlier study using data from the plots we dealt with the relation between leaf age and damage by initial decomposition (Van der Velde & Van der Heijden, 1985). Together with the complementary study about the production of floating leaves (Klok & Van der Velde, 2022) it describes the life and plasticity of floating leaves of Nymphoides peltata under different environmental circumstances. The present study deals with the following questions: (a) What are the causes and patterns of initial decomposition of floating leaves? (b) How does initial decomposition progress during the season and is there a succession of causes?

Materials and Methods

Sites

Research took place in four plots two in outdoor mesocosms and two in an oxbow lake, each with a size of 50 cm × 50 cm (Klok & Van der Velde, 2022) (Table 1).

Table 1 Characteristic data of the plots.

	CT1	CT2	BS1	BS2	
Location	In concrete tank, mixed with Glyceria sp.	In concrete tank, monoculture	Bemmelse Strang, in center of Nymphoides stand	Bemmelse Strang, at open water border of stand	
Depth (cm)	40–50	40–50	30–106	82−154	
Hydrology	Precipitation, evaporation	Precipitation, evaporation	Precipitation, evaporation, river water overflow	Precipitation, evaporation, river water overflow	
Wind and wave action	Low	Low	Moderate	Moderate	
Bottom	Clay, mud	Clay, mud	Sand, detritus	Sand, detritus	
Trophic status	Alkaline/eutrophic	Alkaline/eutrophic	Alkaline/eutrophic	Alkaline/eutrophic	
Chemical characteristics					
Alkalinity (meq.L−1)	−	−	2.6–4.8	2.6–4.8	
pH	−	−	7.6–8.6	7.6–8.6	
Sampling year	1978	1978	1980	1980	
Species	Nymphoides peltata	Nymphoides peltata	Nymphoides peltata	Nymphoides peltata	
	Glyceria fluitans				
	Glyceria maxima				
Note:

CT1, concrete tank 1; CT2, concrete tank 2; BS1, Bemmelse Strang 1; BS2, Bemmelse Strang 2.

Potential, actual and photosynthetic leaf area

A distinction was made between potential, actual and photosynthetic leaf area. The potential area is defined as the area of an entirely intact leaf, the actual area as the potential area minus the area missing and the photosynthetic area as the remaining green part of the actual area.

Regression equation for calculating leaf area

The potential leaf area was calculated from leaf length and leaf width and is described mathematically by Eq. (1), which has been determined by previous research (Van der Velde et al., 1982):

(1) A(L,W)=1.028∗π∗(L+W4)2

where: A(L,W) = potential leaf area at length L and width W (mm2).

L = leaf length (mm).

W = leaf width (mm).

1.028 = correction factor (the leaves are not circular).

Field data

The initial leaf decomposition of N. peltata floating leaf blades was studied in the same plots and at the same time as the leaf production as described in Klok & Van der Velde (2022). Initial decomposition data included visually estimating both leaf damage and decomposition per cause as percentage of the potential leaf area of each leaf. Several types of damage and their causes have been distinguished and described earlier (Lammens & Van der Velde, 1978; Van der Velde, 1979).

Results

Floating leaf data

Floating leaf information, comprising total number of leaves, total potential leaf area, leaf life span, growth period and vegetation period, is shown per plot in Table 2. The combination over time of total number of leaves and total potential leaf area is shown per plot (Fig. 3).

Table 2 Floating leaf information per plot.

Location
Year	CT1
1978	CT2
1978	BS1
1980	BS2
1980	
Total number of leaves	m−2.yr−1	2,552	2,492	1,712	1,112	
Total potential leaf area	m2.m−2.yr−1	3.527	3.82	6.91	5.433	
Leaf life span						
Maximum	d	43	54	59	63	
Minimum	d	3	2	3	3	
Growth period		May 2–Oct 20	May 6–Oct 20	May 18–Oct 20	May 18–Oct 23	
Length	d	171	167	155	158	
Vegetation period		May 2–Nov 20	May 6–Nov 6	May 18–Nov 3	May 18–Nov 3	
Length	d	202	184	169	169	
Note:

CT1, concrete tank 1 (mixed); CT2, concrete tank 2 (mono); BS1, Bemmelse Strang 1 (center); BS2, Bemmelse Strang 2 (border).

Figure 3 Total number of leaves and total potential leaf area in tank plots (CT1, CT2) and lake plots (BS1, BS2).

Additional measurements at the start of the growing season 1979 for CT1 and CT2 are shown in green.

Unfortunately, data of the first weeks in 1978 were missing for CT1 and CT2. Fortunately, the growing season of N. peltata started on the same date both in 1978 and in 1979, so data of the first weeks of 1979 for CT1 and CT2 have been used to give an indication of number of leaves and leaf area at the start of the season.

Causes and patterns of initial decomposition

All causes and stages of initial decomposition found in this study are described below:

Consumption and damage by waterfowl. Consumption of leaf tissue by coots (Fulica atra L., Rallidae) occurred in the lake plots only and can be recognized by missing parts in the form of triangular areas at the margin of leaves. Sometimes major parts of leaves are consumed. Prints of the beak are visible around the consumed areas.

Consumption by pond snails. A major cause of damage on fresh leaf tissue in all plots was caused by Lymnaea stagnalis (L.) (Lymnaeidae, Gastropoda) by consuming parts of young leaves under water, resulting in rows of holes in the unrolled leaf blades, large near the edge and smaller towards the center of the leaf. To a lesser extend other lymnaeid species and other freshwater pulmonate snails were involved, showing a preference for decaying leaf material, in particular areas infected by fungi.

Consumption and damage by aquatic caterpillars of moths. The caterpillars of the moths Elophila nymphaeata (L.), Crambidae, Lepidoptera (brown china-mark, lake plots only) (Gaevskaya, 1969; Lammens & Van der Velde, 1978) and Cataclysta lemnata (L.), Crambidae, Lepidoptera (small china-mark, tank plots only) damaged floating leaves in two ways: by leaf tissue consumption and by cutting out oval leaf patches that are used for shelter (Van der Velde, 1979). Floating shelters are created either by attaching a patch to the underside of a floating leaf or by spinning two patches together (Elophila), or by constructing a floating case by various materials, in particular small leaf pieces (Cataclysta).

Mining by chironomid larvae. Larvae of the midge Cricotopus trifasciatus Mg., Chironomidae, Diptera, were observed to mine their way through the leaf tissue by consuming particular tissue layers while leaving the lower epidermis unaffected (halfminer) (Lammens & Van der Velde, 1978). They occurred in the lake plots only (Fig. 4).

Figure 4 Damage patterns caused by Cricotopus trifasciatus on floating leaves of Nymphoides peltata and floating seed pods of N. peltata.

Infection by phytopathogens. Leaves were infected in all plots by the fungus Septoria villarsiae Desm., Mycosphaerellaceae, Capnodiales, the causative agent of a leaf spot disease (Fig. 5). Asexual spores (conidia) are produced in conidiomata (black spherical structures) (Fig. 6).

Figure 5 Damage spots (leaf spot disease) caused by Septoria villarsiae on a Nymphoides peltata leaf.

Figure 6 Close-up of the damage pattern caused by the ascomycete Septoria villarsiae.

Autolysis. Autolysis occurred in all plots and is visible by the change in leaf color from green to yellow, indicating that chlorophyll is degraded.

Microbial decay. The resistance of a leaf against microbial infection disappears quickly due to erosion of the wax layer and autolysis, facilitating microbial decay in all plots, which was indicated by a change in leaf color from yellow to brown and the softening of the leaf tissue by maceration. During microbial decay, leaves sunk under the water surface.

In tank plots CT1 and CT2 loss and damage by Cataclysta lemnata, pond snails, Septoria villarsiae, autolysis and microbial decay occurred. Bemmelse Strang lake plots BS1 and BS2 suffered from loss and damage by Fulica atra, pond snails, Elophila nymphaeata, Cricotopus trifasciatus, Septoria villarsiae, autolysis and microbial decay. Figure 7 shows the relative contributions to initial decomposition of all causes in all plots.

Figure 7 Relative contributions of all damage causes in tank plots (CT1, CT2) and lake plots (BS1, BS2).

Percentual contributions of number of affected leaves with respect to the total number of leaves in tank plots (CT1, CT2) and lake plots (BS1, BS2). Contributions are listed clockwise, starting at 12 o’clock.

Impact of causes

The impact of initial decomposition causes on leaves for all plots is shown in Table 3. Autolysis and microbial decay are the main decomposition causes. Generally, initial decomposition caused by animals was a very small part of the total potential floating leaf area in all plots, since N. peltata leaves disappear under water rather soon after autolysis and cell death and were thus lost for measurement of further decomposition. The combination of number of leaves and total potential leaf area clearly shows that in the second half of the growth period the tank plots produced smaller leaves, in contrast to the lake plots, as described in Klok & Van der Velde (2022).

Table 3 Impact of damage causes on floating leaves per plot.

Damage cause	Percentage of leaves affected	Potential area affected	Photosynthetic area lost (m2)	
(1)	(2)	(3)	(4)	(1)	(2)	(3)	(4)	(1)	(2)	(3)	(4)	
av	max	av	max	av	max	av	max	
Fulica atra	–	–	15.8	14.8	–	–	–	–	3.92	98	3.25	100	–	–	0.252	0.146	
Snails	15.2	5.14	26.9	13.0	2.91	95	0.99	99	1.51	55	0.58	20	0.095	0.051	0.095	0.031	
Elophila
nymphaeata	–	–	6.8	4.3	–	–	–	–	0.43	25	0.23	30	–	–	0.034	0.011	
Cataclysta
lemnata	0.9	5.5	–	–	0.03	10	0.34	30	–	–	–	–	0.002	0.012	–	–	
Cricotopus trifasciatus	–	–	91.8	93.9	–	–	–	–	17.46	95	19.77	90	–	–	1.162	1.047	
Septoria
villarsiae	68.7	37.6	11.0	10.5	30.77	95	14.83	85	0.35	25	0.24	7	1.244	0.508	0.021	0.016	
Autolysis	86.2	85.1	96.7	43.0	40.74	100	48.49	100	11.28	90	8.26	93	1.704	2.255	0.782	0.463	
Microbial decay	26.0	42.2	92.1	64.6	8.89	90	15.03	85	16.34	98	22.64	97	0.455	0.980	1.322	1.643	
All causes	95.6	91.0	97.0	96.4	84.15	100	84.53	100	23.23	98	23.75	99	3.499	3.806	3.667	3.357	
Note:

The percentage of leaves affected, the average percentage of the potential area affected over all leaves (av) and the maximum potential area affected of a single leaf in mm2 (max) and the total area of lost photosynthetic tissue for all leaves are shown. Where (1), CT1, 1978; (2), CT2, 1978; (3), BS1, 1980; (4), BS2, 1980.

For all plots the absolute and relative loss by damage causes of photosynthetic leaf area over time is shown (Fig. 8) with the impact of damage causes (Table 3).

Figure 8 Absolute and relative damage contributions of causes of initial decomposition in tank plots (CT1, CT2) and lake plots (BS1, BS2).

Per plot the absolute cumulative (above) and the relative (below) contributions are shown.

The order of impact of damage causes from high to low, based on average percentages of the total potential area affected, is per plot: CT1: autolysis, Septoria, microbial decay, snails, Cataclysta,

CT2: autolysis, microbial decay, Septoria, snails, Cataclysta,

BS1: Cricotopus, microbial decay, autolysis, Fulica, snails, Elophila, Septoria,

BS2: microbial decay, Cricotopus, autolysis, Fulica, snails, Septoria, Elophila.

The impact of initial decomposition damage (= loss of photosynthetic leaf area) per cause over time for the plots is shown in Fig. 9 (plot CT1), Fig. 10 (plot CT2), Fig. 11 (plot BS1) and Fig. 12 (plot BS2). All figures use the same scale for proper comparison.

Figure 9 Initial decomposition damage per cause over time for plot CT1 (1978).

Figure 10 Initial decomposition damage per cause over time for plot CT2 (1978).

Figure 11 Initial decomposition damage per cause over time for plot BS1 (1980).

Figure 12 Initial decomposition damage per cause over time for plot BS2 (1980).

The minor damage causes Cataclysta, Elophila, Septoria and snails are also shown with an enlarged Y-scale to display more details (Fig. 13). For Cataclysta, Cricotopus, Elophila and Septoria the damage increments are shown along with the total damage over time (Figs. 14–17). The development of several large generations of midge and moths can be seen (Fig. 9 through Fig. 17). Not all generations exist parallel in time. For Cataclysta, Cricotopus, Elophila and Septoria the summation of the increments is shown for all plots (Fig. 18).

Figure 13 Initial decomposition damage with enlarged Y-scale for Cataclysta, Elophila, Septoria and snails in tank plots (CT1, CT2) and in lake plots (BS1, BS2).

Figure 14 Incremental and total damage of Cataclysta and Septoria in CT1.

Incremental damage is shown left and total damage right.

Figure 15 Incremental and total damage of Cataclysta and Septoria in CT2.

Incremental damage is shown left and total damage right.

Figure 16 Incremental and total damage of some causes in BS1.

Incremental vs total damage of Cricotopus, Elophila and Septoria, with incremental damage shown left and total damage right.

Figure 17 Incremental and total damage of some causes in BS2.

With incremental damage shown left and total damage right.

Figure 18 Summation of incremental damage for Cataclysta and Septoria in CT1 and CT2 and for Cricotopus, Elophila and Septoria in BS1 and BS2.

The percentage of leaves affected by all causes in the tank plots was 91.0% and 95.6%, respectively, which was slightly lower than in the lake plots (96.4% and 97.0%). The average percentage of potential total leaf area affected was stable: high for the tank plots (84.15% and 84.53%) and low for the lake plots (23.23% and 23.75%).

Succession of causes

The succession of damage causes, based on first occurrence, is per plot: CT1 (mixed): autolysis, microbial decay, snails, Septoria, Cataclysta,

CT2 (monospecific): autolysis, microbial decay, snails, Septoria, Cataclysta,

BS1 (center): Cricotopus, microbial decay, autolysis, snails, Elophila, Fulica, Septoria,

BS2 (border): Cricotopus, snails, autolysis, microbial decay, Elophila, Septoria, Fulica.

The above succession lists show that the tank plots (CT1, CT2) have the same succession order, while the lake plots (BS1, BS2) have quite a different order.

Discussion

Without proper statistical analysis (due to the absence of replications), conclusions were made in an observational descriptive way.

Differences between plots

The number of leaves in the tank plots was considerably higher and the size of leaves considerably smaller compared to the lake plots (Table 2) (Klok & Van der Velde, 2022). This can be explained by the limited space and by the limited nutrient availability in the tank plots. At a higher degree of enclosure more and smaller leaves appeared, compared to a low degree of enclosure (Klok & Van der Velde (2022) and literature therein).

Compared to the center plot in the lake, the border plot has a higher nutrient availability through continuous supply via water currents, which counts for fewer, but larger leaves.

The tank plots showed a decrease in leaf area over time after a maximum at the start of the season. Due to the sudden inundation of river water in early spring, the reaching of such a maximum was disrupted in the lake plots. This resulted in a later maximum, which was lower for the center plot due to limited nutrients and much higher for the border plot with much less leaves and almost unlimited nutrients.

The absence of waterfowl in the tank plots was expected, since the plots were covered with a frame with chicken wire. The occurrence of the moth Cataclysta in the tank plots was due to introduction with Lemna in the past, while the moth Elophila occurred in the lake plots. The midge in the lake plots (Cricotopus) probably exists in larger water volumes only where wind and wave action provide the larvae with oxygen. Infection by the fungus (Septoria) was very high in the tank plots and low in the lake plots, which could be caused by the combination of high leaf density and low nutrient availability in the tank plots.

The average percentage of potential leaf area affected, high for the tank plots and low for the lake plots, was caused by the very low infection by Septoria in the lake plots compared to the tank plots and possibly by the high leaf density and the low nutrient availability in the tank plots. Snails had a large impact in one tank plot.

Incremental damage was higher for CT1 and BS1 compared to CT2 and BS2, respectively, except for Cataclysta.

Senescence

Newly unrolled floating leaf blades are fully green and hydrophobic due to a thick epicuticular wax layer. This layer gradually erodes during senescence and as cellulolytic and other bacteria and fungi colonize the leaf tissue (Howard-Williams, Davies & Cross, 1978; Robb et al., 1979; Rogers & Breen, 1981; Barnabas, 1992). Senescence starts shortly after the first leaves are fully grown and continues throughout the growth period. During senescence the leaves turn from green to yellow by autolysis, an orderly physiological process controlled by the plant itself, and ultimately turn to brown. Concomitant microbial decay softens the leaves.

Van der Velde & Van der Heijden (1985) analyzed the relative increase of different damage and decomposition types at different leaf age classes (each class 5 days) of the floating leaves of N. peltata in the same plots as used in this study. They compared the results between the concrete tanks (two plots together) and the Bemmelse Strang (two plots together). Patterns of occurrence of the decomposition types over leaf age classes shows clearly that young leaves are damaged already when they appear at the water surface in the case of Cataclysta lemnata, (age 1–9, peak at age 3) and Elophila nymphaeata (age 1–7, peak at age 1), Cricotopus trifasciatus (age 1–13, peak at age 4) snails (age 1–9 CT, peak at age 6 and BS age 1–9, peak at age 1) and Coots (Fulica atra) (age 1–10, peak at age 2). Septoria villarsiae started in the concrete tanks at leaf age 1 with a peak at age 4–5 (leaf age –11), while in the Bemmelse Strang it started at leaf age 4 with a peak at age 11 (leaf age 4–13). Yellow areas started in both waters at age 1 with peak at 5 (CT 1-7, BS 1-9). Decayed areas replaced the yellow ones at leaf age 1–11 with a peak at age 8 in the case of the concrete tanks and at leaf age 1–13 with a peak at age 12 in the case of the Bemmelse Strang. Brock et al. (1983) investigated the nitrogen and phosphorus concentrations (μmol per g dry weight) in N. peltata leaves in the Bemmelse Strang. Young green leaves showed the highest N concentration (2,090 μmol per g dry weight), mature green leaves (1,275 μmol per g dry weight), mature leaves (1,288 μmol per g dry weight), senescent leaves (1,281 μmol per g dry weight) and decaying leaves higher concentrations (1,363 μmol per g dry weight). The P concentrations showed a decrease with the highest values in P of the young green leaves (168 μmol per g dry weight) and 93, 90, 74 and 66 μmol per g dry weight for the other stadia respectively.

From these results we can conclude that the specialist herbivores prefer young leaves as food. Yellow and decayed areas showed a similar course and peaked after each other. The clearest difference was the timing of Septoria infection, which started early in young leaves in the concrete tanks but in the Bemmelse Strang in older leaves. This can be explained by the much better nutrient situation in the Bemmelse Strang compared to that in the concrete tanks where nutrients are a limiting factor leading to smaller leaves with a shorter leaf life span (Klok & Van der Velde, 2022).

Succession of causes

The succession of leaf decomposition causes, based on first occurrence, was exactly the same for both tank plots: autolysis was followed by microbial decay, snails, Septoria and finally Cataclysta.

For the lake plots this succession was different. Both plots started with Cricotopus, for the center plot followed by microbial decay, autolysis, snails, Elophila, Fulica and Septoria, and for the border plot followed by snails, autolysis, microbial decay, Elophila, Septoria and Fulica.

Primary underwater consumption of young leaves by snails may explain the different succession order in the lake plots for snails, autolysis and microbial decay. The late occurrence of microbial decay in the border plot may be explained by more available nutrients strengthening the leaf condition.

Influences on susceptibility

Prolonged cloudy and wet weather are supposed to impose stress on Nymphoides by weakening the defense of leaves due to reduced solar radiation, and thus promoting heavy infection and damage by phytopathogens as demonstrated for floating leaf blades of Nymphaea alba (Van der Aa, 1978). Leaf blades contain phenolics with fungistatic properties (Smolders et al., 2000) and poor light conditions reduce the content of phenolics in leaf tissue as demonstrated for floating leaf blades of waterlilies (Vergeer & Van der Velde, 1997), which makes mature leaves vulnerable to infection by fungi and oomycetes.

Comparison with other nymphaeids

The growth period of N. peltata in the plots lasted 155 to 171 days, or 85% to 93% of the vegetation period of 169 to 202 days of the plots. The total loss of photosynthetic area of N. peltata was 99.9% for CT1, 100.0% for CT2, 53.1% for BS1 and 60.5% for BS2 (Tables 2 and 3).

From similar research on plots with Nuphar lutea, Nymphaea alba and N. candida (Klok & Van der Velde, 2017, 2019), the growth period lasted 71 to 134 days, or 53% to 73% of the vegetation period of 135 to 199 days. Percentages of loss of photosynthetic area ranged from 38% to 50% for Nuphar and 44% to 54% for Nymphaea.

The relatively long growing period for N. peltata indicates that it has characteristics of an invader (Tippery et al., 2023) with continuous production of floating leaves deriving from runners by which the plant can easily colonize and expand its area very fast and so recover quickly from disturbance in contrast to the waterlilies studied with their massive rhizomes and long term colonization and dominance.

Conclusions

Initial decomposition included the internal cause autolysis and external causes by animals, fungi and microbes. Animals included birds (Fulica atra), snails (mainly Lymnaea stagnalis), caterpillars of aquatic moths (Cataclysta lemnata, Elophila nymphaeata) and chironomid larvae (Cricotopus trifasciatus). The ascomycete Septoria villarsiae was the most important fungus. Autolysis and microbial decay were the main causes, while the other causes generally were marginal.

From high to low impact the order of initial decomposition causes was autolysis, Septoria, microbial decay, pond snails, Cataclysta for the tank plots and Cricotopus, microbial decay, autolysis, Fulica, pond snails, Elophila, Septoria for the lake plots.

The succession of causes per plot, based on first occurrence, was autolysis, microbial decay, snails, Septoria, Cataclysta for the tank plots and it was Cricotopus, microbial decay, autolysis, snails, Elophila, Fulica, Septoria for the center lake plot and Cricotopus, snails, autolysis, microbial decay, Elophila, Septoria, Fulica for the border lake plot.

Supplemental Information

Supplemental Information 1 Tank 1 initial decomposition of floating leaf blades of Nymphoides peltata (S.G. Gmel.) O. Kuntze: causes, impacts and succession.

Click here for additional data file.

Supplemental Information 2 Tank 2 Initial decomposition of floating leaf blades of Nymphoides peltata (S.G. Gmel.) O. Kuntze: causes, impacts and succession.

Click here for additional data file.

Supplemental Information 3 Bemmelse Strang 1 Initial decomposition of floating leaf blades of Nymphoides peltata (S.G. Gmel.) O. Kuntze: causes, impacts and succession.

Click here for additional data file.

Supplemental Information 4 Bemmelse Strang 2 Initial decomposition of floating leaf blades of Nymphoides peltata (S.G. Gmel.) O. Kuntze: causes, impacts and succession.

Click here for additional data file.

We thank L. van der Heijden, P. M. M. Bexkens and P. A. J. van Grunsven for collecting and working out field data, E. L. Huijser (†) for making the field data digitally available, P. H. M. Charpentier for providing us with all necessary literature and O. Knoppers for checking the English language and editor Curtis Daehler and two anonymous reviewers for their constructive remarks on the manuscript which led to improvement of the article.

Additional Information and Declarations

Competing Interests

Author Contributions

Data Availability

The authors declare that they have no competing interests.

Peter F. Klok conceived and designed the experiments, performed the experiments, analyzed the data, prepared figures and/or tables, authored or reviewed drafts of the article, and approved the final draft.

Gerard van der Velde conceived and designed the experiments, performed the experiments, analyzed the data, prepared figures and/or tables, authored or reviewed drafts of the article, and approved the final draft.

The following information was supplied regarding data availability:

The collected field and mesocosm data are available in the Supplemental Files.

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
