# Peer review of "Initial decomposition of floating leaf blades of Nymphoides peltata (S.G. Gmel.) O. Kuntze (Menyanthaceae): causes, impact and succession"

_PeerJ, doi:10.7717/peerj.16689_

## Round 0.1 · original submission · Major Revisions

The reviewers have suggested various revisions to improve the manuscript, including changes to the Introduction and presentation of the Results and Discussion. The manuscript would also benefit from a clearer statement of motivation for the study and / or a statement or need or knowledge gap addressed, as well as presentation of specific hypotheses to be addressed. Note that reviewer #1 typed specific comments on the PDF manuscript (to be downloaded from the editorial system) in addition to providing a written summary review.

Reviewer 1 ·

Basic reporting

A thorough literature review was provided in the introduction, but I struggled to find a clear motivation for the study. Although the paper does not necessarily need to have a high impact, some tweaking of the introduction could provide motivation for why the study was conducted. The background literature is there, but it is not used to make a strong case for the study. Some statements are made without citing literature where it seems necessary (e.g., lines 69 to 72). Similarly, with some adjustments, the introduction could link ideas better to provide a strong context for the study. In some cases, some information does not seem relevant to the study (e.g., line 116).

The figures presented are in-depth and informative, but it might be easier for readers to digest if it is possible for them to be summarised or presented in such a way as to present the data most relevant to the study. The raw data was shared and is understandable, although it is not clear what the last column (coloured and numbered) represents in three of the four datasheets.

Although the authors may not be mother-tongue speakers of the English language (and if this is the case, they should be commended on their ability to write a paper in another language at all), the grammar used throughout the paper could be improved in several ways. Sometimes, the wording is not clear, and the word order is not quite correct. In some sections, past and present text is used alternately, when past tense would be more appropriate. Commas could be used in some areas where they are not, though this may not be considered to be as important. Assistance from a fluent English speaker or editor could be helpful in this regard.

Experimental design

The research objectives were listed clearly, but it was not stated how they fill an identified knowledge gap. The described methods did not mention sample size or replication, though the datasheets show that numerous leaves were sampled, and some sections were not clear (for example, lines 192 to 196 of the results section discusses the years in which the study was conducted, but there was no mention of this in the methods). In some cases, the information was found in the provided tables, but specifically mentioning the information in the text would make comprehension of the paper much easier for readers.

However, the tanks and oxbow lake used for the study were well-described, and the methods were reproducible, with clear definitions of how photosynthetic area and leaf area were calculated.

Validity of the findings

The datasheet provided suggests that there is an abundance of data to work with to answer the research questions provided, but not all of this was utilised. The first three research questions were answered to an extent, but the fourth question ("Which factors make the leaves susceptible for decomposition?") seemed neglected. For example, temperatures were recorded in the datasheet, but these were not related to decomposition rate or type, whereas this could be a useful relationship to investigate, especially since figures of decomposition over time were provided for each cause. Seasonal effects could also be important to disentangle effects of season on leaf area measurements.

There is much room for improvement in the discussion section of the paper. Many sections describe the data without discussing it, and should be put in the results section (e.g., lines 192-195 and 295-298). Some literature is cited and discussed, but it is not clearly linked to the research study, and there is room for more references to be discussed and added. The conclusion section summarises the findings of the paper, but could be improved by linking the results to a broader context (i.e., reinforcing the purpose of the research).

Additional comments

I have attached a pdf with some suggestions for improvement. I think there is a lot of potential for this paper based on the data that is there, and some adjustments could greatly improve its quality.

Annotated reviews are not available for download in order to protect the identity of reviewers who chose to remain anonymous.

Reviewer 2 ·

Basic reporting

I was very excited to read the article entitled “Initial Decomposition of Floating Sheets of Nymphoides peltata (S.G. Gmel.) O. Kuntze: Causes, Impacts and Succession” written Peter Klok, Gerard van der Velde. The article was well written and interesting. The article included an excellent background on the decomposition of aquatic macrophytes in the introduction. The article was very well structured with many tables and figures about the results found. It was possible to visualize very well the patterns of decomposition of the focal species of the study. Unfortunately, I did not find premises or hypotheses that anchored the study, instead, the authors chose to seek to answer three relevant questions. I strongly encourage authors to try to include premises and hypotheses that further reinforce the framework laid out in the introduction.

Experimental design

The experimental design of the article was very well written and detailed. Despite this, it was not described how many replicates there were for each treatment and in each location used in the study. For better visualization, a visual representation of the experimental design would be good. In addition, it would be relevant to describe more clearly in the text the experimental period in each location used in the study.

Validity of the findings

Although the study demonstrated the causes and patterns of initial decomposition of floating leaves very well with numerous graphs and tables, to me the biggest flaw is the lack of statistical analysis of the data comparing treatments and study sites, as well as a way to reinforce the results of successive causes, patterns and processes involving the initial decomposition of floating leaves. I would strongly suggest carrying out these statistical analyzes in order to better anchor the conclusions found.

Additional comments

lines 53-54: The statement: "The number of studies on nymphaeids is few compared to those of the latter two groups." does not make it very clear in the context of the paragraph which these two groups are. Please rewrite.

Figure 7: Write the contribution percentage of each damage cause on the pie chart.

---

## Round 0.2 · Minor Revisions

The two reviewers had opposing opinions, one recommending "minor revisions" and the other recommending "reject". The reject recommendation was based on lack of statistical comparisons. The authors and the other reviewer (#1) also recognize this issue, but they make a case that the data are interesting and still worth publishing, and I think this is reasonable. Nevertheless, wording currently presented in the manuscript regarding lack of statistical testing needs to be revised to improve clarity. Reviewer #1 has suggested specific wording. Furthermore, the following two issues also need to be addressed in a revised manuscript:

1. The second author published a previous paper in 1985 that sounds very similar to the present manuscript: Van der Velde G, Van der Heijden LA. 1985. Initial decomposition of floating leaves of Nymphoides peltata (Gmel.) O. Kuntze (Menyanthaceae) in relation to their age, with special attention to the role of herbivores. Verhandlungen der Internationalen Vereinigung für theoretische und angewandte Limnologie 22:2937-2941. The authors should explain in their Introduction what was found in this 1985 study and what the present manuscript contributes beyond what was reported in 1985.

2. The fourth research question "Which factors make the leaves susceptible for decomposition?" was not addressed at all by the study at hand, so I would ask that the author remove this question from their Introduction.

I also have various additional comments and edits that I typed on the manuscript (attached as PDF). I will also ask PeerJ staff to send this file to you in Word format since that may be easier for you to work from as you prepare a revised manuscript.

Reviewer 1 ·

Basic reporting

The manuscript is much improved from the last draft. Some sections were written in present tense and should be changed for consistency.

Experimental design

The description of experimental design is sufficient.

Validity of the findings

Although there was not enough replication to conduct statistical analyses, as a qualitative observational study, the results are interesting. I have made a suggested change to the sentence in the discussion that refers to the lack of statistical stidies.

Additional comments

The manuscript is much clearer than the first draft and should be accepted after some minor revisions. I have made comments in the attached PDF for perusal.

Annotated reviews are not available for download in order to protect the identity of reviewers who chose to remain anonymous.

Reviewer 2 ·

Basic reporting

Thank you very much for sending this new version of the article. The article is really well written and quite interesting. I reiterate the valuable introduction, with an excellent history on the decomposition of aquatic macrophytes. Although it is possible to visualize the decomposition patterns of the focal species of the study, without statistical analyzes that confirm these patterns, it is unfortunately difficult to have a robust interpretation of the information arising from the results. As described by the authors, the experimental design did not allow replicates, which does not make statistical analyzes possible.Therefore, I believe that the article is unfortunately not suitable for publication in the PeerJ Journal. I strongly encourage authors to revise the experimental design to try to resolve this issue for future submissions.

Experimental design

As described by the authors at the beginning of the discussion of this new version "Unfortunately, due to the fact that the four locations did not contain multiple plots, it is useless to apply statistics for firm conclusions.", the absence of statistics weakens the robustness of the results found.

Validity of the findings

Without proper statistical analysis (product of the absence of replications), conclusions end up being made in an observational, subjective way and with only descriptive results.

---

## Round 0.3 · accepted · Accept

The authors have addressed the reviewer's comments, and I recommend the manuscript for publication; however, I recommend a few minor edits as follows:
L 23 “simultaneously by measuring” to “by simultaneously measuring”
L 97 “question:” to “questions:”
L 206 “was91.0” to “was 91.0”
L 221 “product of” to “due to”
L 274-279 – Please add units for these measurements
L 318 “restore” to “recover”
L 344 “lead” to “led”